# Efficient Experimentation for Estimation of Continuous and Discrete Conditional Treatment Effects

**Muhammed T. Razzak** *
OATML Group
University of Oxford

**Panos Tigas** *
OATML Group
University of Oxford

**Andrew Jesson**
Blei Lab
Columbia University

**Yarin Gal**
OATML Group
University of Oxford

**Uri Shalit**
Technion - Israel Institute of Technology

## Abstract

Accurately estimating personalized treatment effects often demands substantial data, incurring high costs across diverse applications such as personalized advertisement delivery and clinical trials. Existing methodologies employ deep models to estimate treatment effects in high-dimensional data, often relying on randomly selected experiments. We explore the potential of active learning techniques to enhance the efficiency of experimentation. Our focus centers on a relatively underexplored yet common scenario where each unit is subject to experimentation only once. We build upon the Bayesian active learning framework, to select units, and a treatment to apply to the unit, that maximize the information gain from each experiment. Our approach is flexible, accommodating both discrete and continuous treatment settings. Furthermore, we address the inefficiencies in batch experimentation by employing a greedy and a policy gradient-based optimization strategy. We validate the effectiveness of our proposed method on synthetic and high-dimensional semi-synthetic datasets (based on IHDP and TCGA). Our results show significant improvements in experimentation efficiency over the baseline methods.

## 1 Introduction

Experimentation often incurs significant costs in terms of resources, time, and funding. This necessitates the development of more efficient and targeted experimental designs. Adaptive trial design is a promising paradigm for conducting experiments, allowing more efficient use of scarce resources (Foster et al., 2020). Given a pool of samples and a set of (possibly continuous) interventions or treatments we wish to study, our goal is to construct an experiment that yields the most informative results about the effects of the interventions on a predefined outcome, while minimizing costs and resources used.

The majority of experimental design literature has primarily focused on a setting in which one may conduct experiments on the same object multiple times, usually with differing actions (also known as treatments, or interventions) (Rainforth, 2017). A somewhat less explored scenario is where one can actively select a unit from a pool of available units, and then select only a *single* action to apply to this unit. This scenario has been referred to as active learning for trials (Deng et al., 2011) or adaptive trials (Chen et al., 2021). This scenario is relevant in clinical trials, animal studies, and any other

Workshop on Bayesian Decision-making and Uncertainty, 38th Conference on Neural Information Processing Systems (NeurIPS 2024).

scenario where the units are heterogeneous and can change after being subjected to the intervention, or can only be treated once for ethical or cost reasons.

Our paper makes several contributions to the field of active learning for experimentation:

1. We propose a new setting for active learning in which the objective is to efficiently select the most informative unit-treatment combinations for experimentation (Section 2).
2. We utilize deep learning methods enabling us to model units with high-dimensional co-variates and explore a wide range of treatment options (binary, discrete, and continuous options). We select unit-treatment combinations with the highest Expected Information Gain by introducing both a greedy algorithm and novel gradient-based algorithm to optimize the joint mutual information between all experiments selected in each batch (Section 3).
3. We demonstrate the efficacy of our methods on synthetic and high-dimensional semi-synthetic datasets, resulting in significant improvements in efficiency (Section 4).

## 2   Problem Formulation

Within the Neyman-Rubin causal model (Rubin, 1974), each unit $u$ has a potential outcome $Y^t(u)$ associated with every intervention $t \in \mathcal{T}$, where $\mathcal{T}$ is the (possibly infinite) set of interventions under consideration. We are interested in efficiently learning the conditional average potential outcome (CAPO) function, $\mathbb{E}[Y^t \mid \mathbf{X} = \mathbf{x}]$. In the regime of continuous interventions, this is often referred to as the (conditional) "dose response function."

The expectation of the potential outcome given a set of covariates $\mathbf{X} = \mathbf{x}$ describing $u$ is identifiable from data where $T$ is directly intervened on, assuming that $\mathbf{X}$ and $T$ are causal parents of $Y$, that the observed outcome corresponds to the potential outcome of the assigned treatment, and that the treatment assigned to one unit, $u$, does not effect the outcome observed for any other unit, $u'$. Under these conditions, the CAPO takes the form of the expectation

$$\mu(\mathbf{x}, t) := \mathbb{E}\left[Y \mid \mathbf{X} = \mathbf{x}, T = t\right]. \tag{1}$$

We are specifically interested in a $k$-round, hybrid, "pool based" setting, consisting of a non-replenishable pool of units, $U = \{u_i\}_{i=1}^n \subseteq \mathcal{U}$, and a set of possible interventions $T \subseteq \mathcal{T}$. This is a common setting, where experimenters will do sequential experiments either in batch or non-batch setting and have access to each unit once. A common example is preclinical trials, done in stages on animals to study safety, efficacy, and biological activity of a drug. Each unit, described by high-dimensional covariates, is unique and non-replenishable.

We consider both discrete and continuous intervention spaces, $\mathcal{T}$. At each round $i \in [k]$, the experimenter uses a policy, $\pi : \mathcal{U} \times \mathcal{T} \to \mathcal{S} \subseteq \mathcal{X} \times \mathcal{T}$, to select a $b$-sized batch of unit-intervention pairs, $s = \{\mathbf{X}(u_j) = \mathbf{x}_j, t_j\}_{i=j}^b$. The experiments are then run and the experimenter observes the outcomes, $\{y_j\}_{j=1}^b$, for each.

At the end of each round, the experimenter fits an estimator, $\mu(\mathbf{x}, t; \boldsymbol{\theta})$, of the CAPO, $\mu(\mathbf{x}, t)$, to the data accumulated, $\mathcal{D}_{\text{train}} = \cup_{i=1}^k \left\{ \{\mathbf{x}_j, t_j, y_j\}_{j=1}^b \right\}_i$.

## 3   Method

This section outlines our proposed selection policies which aim to maximize the Expected Information Gain over a set of experiment designs in a batch of size $b$, denoted by $\xi_{i=1:b} = (\mathbf{x}_i, t_i)$.

In contrast to traditional experimental design, we need to acquire units, $\mathbf{x} \in U$, **without replacement** and select the treatment, $t \in \mathcal{T}$, **with replacement**. Furthermore, while units are discrete and drawn from a pool set, treatments can be discrete or continuous.

The problem is defined as selecting a batch of experiments that maximize the EIG objective:

$$\{(\mathbf{x}, t)_{1:b}\}^* = \arg\max_{\{(\mathbf{x}, t)_{1:b}\}} \text{EIG}(\{(\mathbf{x}, t)_{1:b}\}) \tag{2}$$

where the utility set function is defined as the EIG objective $\text{EIG}(\cdot) \triangleq I(Y; \boldsymbol{\Theta} \mid \cdot, \mathcal{D}_{\text{pool}})$.

We propose several acquisition policies for both discrete and continuous treatment settings:

**Top-K.** This strategy selects the top-k design with the highest estimated EIG per design. However, this does not maximize the EIG of the batch, as it does not take into account how other experiments in the batch affect the mutual information of each individual score. This can result in redundant experiments being conducted and results in inefficient experimentation (Foster et al., 2021; Kirsch et al., 2019).

**Softmax Top-K.** Kirsch et al. (2022) note this inefficiency and propose sampling the top-k experiments from a Softmax distribution, with temperature $\beta$, to the EIG scores. This selection policy has shown superior performance in terms of efficiency. It is hypothesized that this is due to the perturbation of the distribution taking into account that the mutual information changes when other experiments are selected.

**Greedy Optimization.** When the set function we wish to optimize is known to be submodular and non-monotonic, then a greedy optimization-based algorithm can be used to maximize the set, which enjoys $1 - \frac{1}{e}$ approximation guarantees (Nemhauser et al., 1978). As shown in (Tigas et al., 2022; Kirsch et al., 2019), Expected Information Gain over continuous outcomes (i.e. regression problems) is both submodular and non-monotonic thus it is suitable for applying a Greedy optimization-based algorithm.

**Policy-Gradient Based Optimization.** To alleviate the approximation shortcomings of the greedy optimization-based algorithm, we notice that the objectives are differentiable with respect to the trial parameters (units and dosages), thus we can employ gradient-based methods to design experiments that maximize our objectives. We parametrize the policy $\pi_\phi(X, T)$ as joint distribution over units $X$ and dosages $T$.

We provide a more details on each method in Appendix B.

## 4 Experiments

We evaluate our acquisition policies on synthetic and semi-synthetic datasets, and show significant improvements over the baseline.

**Datasets.** We evaluate on 3 datasets: a synthetic dataset, and two semi-synthetic datasets based on the features/covariates in the IHDP (Hill, 2011; Shalit et al., 2017) and TCGA datasets (Cancer Genome Atlas Research Network et al., 2013). For the **synthetic** dataset, suitable for discrete and continuous treatments, we develop a dose-response function based on the generalized dose response function developed in Taleb & West (2023). The covariates of the units are 1-dimensional and normally distributed, with the treatment being either a continuous range or discrete values. For the two semi-synthetic datasets, we obtain covariates from two standard benchmark datasets in causal inference, IHDP and TCGA. The units in **IHDP** dataset contain 25 features, while the units in the **TCGA** dataset contain 4000 features. Further details are provided in Appendix C.

**Model.** Our objectives rely on methods that are capable of modelling uncertainty and handling high-dimensional data modalities. For this we rely upon Deep Bayesian Neural Networks (BNNs). We utilize Deep Ensembles (Lakshminarayanan et al., 2017) which can be seen as approximate Bayesian Inference (Wilson & Izmailov, 2020). We utilize a simple multi-layer perceptron S-learner (Rumelhart et al., 1986; Künzel et al., 2019). We provide the hyperparameters in Appendix D.

**Metrics.** The Mean Integrated Square Error (MISE) measures how well the model estimates the conditional average potential outcome (CAPO) across the entire dosage space:

$$\text{MISE} = \frac{1}{N} \frac{1}{k} \sum_{\mathrm{u} \in \mathrm{U}} \sum_{i=1}^{N} \int_{\mathcal{D}_{\mathrm{t}}} \left( y^i(\mathrm{u}, t) - \hat{y}^i(\mathrm{u}, t) \right)^2 \mathrm{d}t. \tag{3}$$

The metric is computed over a held-out test-set. This metric reflects our objective: to efficiently build an understanding of the entire unit-treatment-response function, through as few experiments as necessary.

Table 1: MISE for proposed acquisition functions on the continuous synthetic dataset, for various acquisition sizes and number of experiments conducted.

| Acquisition Size | 4 | | | 8 | | | 16 | | |
|---|---|---|---|---|---|---|---|---|---|
| Experiments Conducted | 26 | 34 | 42 | 26 | 42 | 74 | 26 | 42 | 74 |
| Random | 0.2786 ± 0.0268 | 0.1814 ± 0.0125 | 0.1519 ± 0.0109 | 0.2651 ± 0.0055 | 0.1794 ± 0.0156 | 0.1482 ± 0.0082 | 0.259 ± 0.0125 | 0.1802 ± 0.0161 | 0.1416 ± 0.0108 |
| Soft Top-K | 0.2727 ± 0.0427 | 0.0393 ± 0.0159 | **0.0354 ± 0.0132** | 0.3139 ± 0.1524 | 0.0685 ± 0.0289 | **0.0142 ± 0.0091** | 0.4877 ± 0.1829 | 0.1503 ± 0.0976 | 0.0167 ± 0.0113 |
| Greedy | 0.1922 ± 0.0601 | 0.0727 ± 0.0166 | 0.0487 ± 0.0265 | **0.1433 ± 0.0737** | 0.0841 ± 0.0372 | 0.0261 ± 0.0112 | **0.1137 ± 0.0532** | 0.1075 ± 0.0387 | 0.0159 ± 0.0079 |
| Policy | **0.013 ± 0.0032** | **0.0077 ± 0.0051** | 0.0456 ± 0.0109 | 0.2268 ± 0.1666 | **0.0322 ± 0.0021** | 0.0473 ± 0.0336 | 0.2306 ± 0.1397 | **0.0071 ± 0.0003** | **0.0061 ± 0.0025** |

Table 2: MISE for proposed acquisition functions on the IHDP dataset, for various acquisition sizes and number of experiments conducted.

| Acquisition Size | 4 | | | 8 | | | 16 | | |
|---|---|---|---|---|---|---|---|---|---|
| Experiments Conducted | 64 | 128 | 192 | 64 | 128 | 192 | 64 | 128 | 192 |
| Random | 0.2786 ± 0.0268 | 0.1814 ± 0.0125 | 0.1519 ± 0.0109 | 0.2651 ± 0.0055 | 0.1794 ± 0.0156 | 0.1482 ± 0.0082 | **0.259 ± 0.0125** | 0.1802 ± 0.0161 | 0.1416 ± 0.0108 |
| Soft Top-K | 0.2727 ± 0.0427 | 0.1895 ± 0.0185 | 0.1369 ± 0.0146 | 0.2784 ± 0.0314 | 0.1615 ± 0.0074 | 0.1233 ± 0.002 | 0.3241 ± 0.0393 | 0.1626 ± 0.0073 | 0.1271 ± 0.0096 |
| Greedy | 0.2814 ± 0.0367 | **0.1434 ± 0.0125** | **0.1107 ± 0.0055** | 0.2444 ± 0.0149 | 0.1456 ± 0.0075 | **0.1204 ± 0.0091** | 0.2755 ± 0.0186 | 0.169 ± 0.0203 | 0.1296 ± 0.0113 |
| Policy | **0.25 ± 0.0138** | 0.1531 ± 0.0107 | 0.1393 ± 0.0042 | **0.2023 ± 0.0043** | **0.1389 ± 0.0088** | 0.1274 ± 0.0167 | 0.2682 ± 0.0054 | **0.1446 ± 0.0062** | **0.1264 ± 0.0083** |

**Results.**  Table 1 shows the MISE for proposed acquisition functions on the continuous synthetic dataset, for various acquisition sizes and number of experiments conducted. We provide further discussion and results in Appendix A.

Our experiments demonstrate several key findings:

**EIG optimization Enhances Covariate Coverage and Counterfactual Estimation.**  We investigate how designing experiments for high EIG changes the covariate and treatment training density, on the discrete synthetic dataset. As illustrated in Figure 1 (in A.1), the use of the EIG objective resulted in a more strategic distribution of the covariate space, with a higher density in regions where the outcome is more uncertain or variable. In the treatment space, we find a level of overlap that allows for more accurate estimation of the counterfactuals (Jesson et al., 2021). This improved coverage and estimation leads to a reduction in MISE, as evidenced by the more accurate covariate-response curves shown in the figure.

**Synthetic Dataset Performance.**  The Greedy and Soft Top-K acquisition functions demonstrate strong performance on the discrete synthetic dataset. Notably, the Greedy method consistently achieves the lowest MISE values across various acquisition sizes and numbers of experiments conducted, making it the best-performing method for discrete treatments. When comparing the performance on the continuous synthetic dataset to the discrete synthetic dataset, we observe that the MISE values are generally lower for the discrete case. The policy gradient method begins to outperform the other acquisition functions as the acquisition size increases.

**Policy-gradient offers robust and scalable performance on high-dimensional data.**  We evaluate our acquisition functions' performance on two semi-synthetic datasets: IHDP (Table 3 and TCGA (Table 4 in Appendix A). These datasets are designed to test the efficiency of our methods in high-dimensional settings, the primary focus of our work. The Greedy and Policy methods consistently achieve the lowest MISE values. For instance, on the IHDP dataset, Greedy achieves a MISE value of 0.1107 for the smallest acquisition size (4) when conducting 192 experiments, outperforming the random baseline by **27.1%**. The Policy method performs well, particularly for larger acquisition sizes and a higher number of experiments conducted.

The results on these datasets demonstrate that our methods and acquisition functions are scalable, versatile, and effective in real-world scenarios. The greedy and policy-gradient acquisition functions, in combination with deep learning for effective representation learning, provide an effective framework for efficient experimentation and estimation of conditional treatment effects.

# 5   Conclusion

In this paper, we have proposed a novel framework for efficiently estimating treatment effects. We provided theoretical justification for our information-theoretic based objective functions, along with our combinatorial optimization. We demonstrated our methods on both synthetic and semi-synthetic datasets, showing significant gains in efficiency over the standard baseline. Future work could include extending our methods to handle larger batch sizes, and incorporating additional objectives for additional utility.

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

# A   Additional Results

## A.1   Detailed Analysis of EIG Optimization Effects

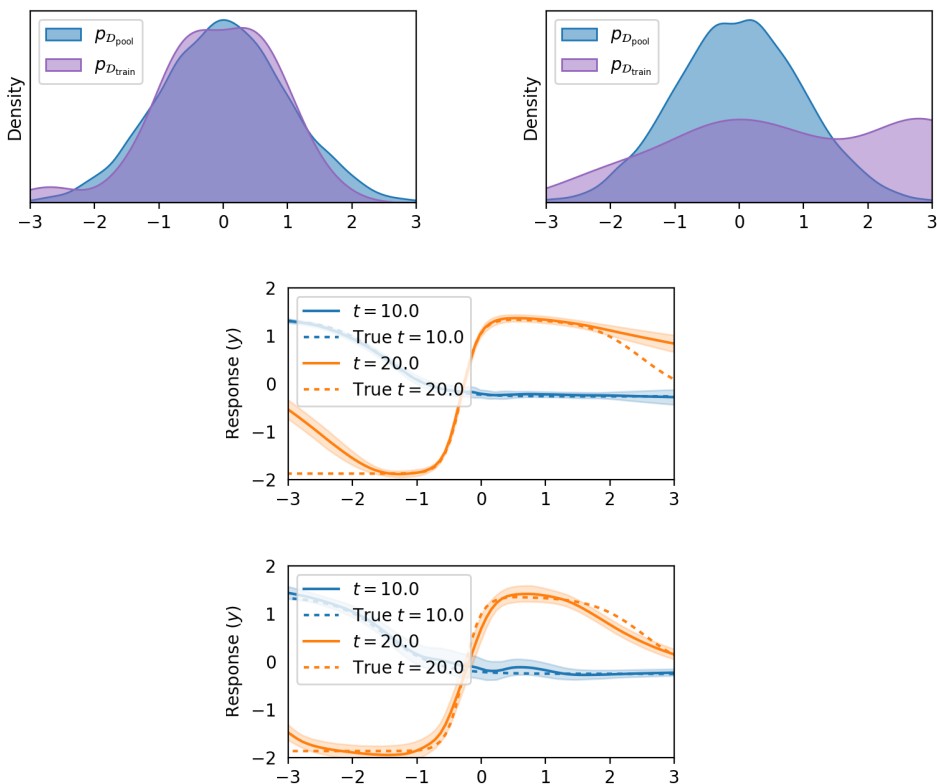

Figure 1: Comparison of data density (top) and covariate response (bottom) using random selection (left) and our method (Soft Top-K) (right) after 80 points were acquired, on our synthetic dataset, with discrete treatments. Our method results in a more evenly distributed point density and a more accurate covariate-response curve.

## A.2   IHDP Dataset Results

Table 3 shows the MISE for proposed acquisition functions on the IHDP dataset, for various acquisition sizes and number of experiments conducted.

Table 3: MISE for proposed acquisition functions on the IHDP dataset, for various acquisition sizes and number of experiments conducted.

| Acquisition Size | 4 | | | 8 | | | 16 | | |
|---|---|---|---|---|---|---|---|---|---|
| Experiments Conducted | 64 | 128 | 192 | 64 | 128 | 192 | 64 | 128 | 192 |
| Random | 0.2786 ± 0.0268 | 0.1814 ± 0.0125 | 0.1519 ± 0.0109 | 0.2651 ± 0.0055 | 0.1794 ± 0.0156 | 0.1482 ± 0.0082 | **0.259 ± 0.0125** | 0.1802 ± 0.0161 | 0.1416 ± 0.0108 |
| Soft Top-K | 0.2727 ± 0.0427 | 0.1895 ± 0.0185 | 0.1369 ± 0.0146 | 0.2784 ± 0.0314 | 0.1615 ± 0.0074 | 0.1233 ± 0.002 | 0.3241 ± 0.0393 | 0.1626 ± 0.0073 | 0.1271 ± 0.0096 |
| Greedy | 0.2814 ± 0.0367 | **0.1434 ± 0.0125** | **0.1107 ± 0.0055** | 0.2444 ± 0.0149 | 0.1456 ± 0.0075 | **0.1204 ± 0.0091** | 0.2755 ± 0.0186 | 0.169 ± 0.0203 | 0.1296 ± 0.0113 |
| Policy | **0.25 ± 0.0138** | 0.1531 ± 0.0107 | 0.1393 ± 0.0042 | **0.2023 ± 0.0043** | **0.1389 ± 0.0088** | 0.1274 ± 0.0167 | 0.2682 ± 0.0054 | **0.1446 ± 0.0062** | **0.1264 ± 0.0083** |

### A.3 TCGA Dataset Results

Table 4 shows the MISE for proposed acquisition functions on the TCGA dataset, for various acquisition sizes and number of experiments conducted.

Table 4: MISE for proposed acquisition functions on the TCGA dataset, for various acquisition sizes and number of experiments conducted.

| Acquisition Size | 4 | | | 8 | | | 16 | | |
|---|---|---|---|---|---|---|---|---|---|
| Experiments Conducted | 64 | 128 | 192 | 64 | 128 | 192 | 64 | 128 | 192 |
| Random | $0.2768 \pm 0.032$ | $0.1968 \pm 0.0154$ | $0.1701 \pm 0.0166$ | $0.2778 \pm 0.0206$ | $0.1988 \pm 0.0182$ | $0.1734 \pm 0.0067$ | $0.3019 \pm 0.0252$ | $0.2058 \pm 0.0099$ | $0.1523 \pm 0.0049$ |
| Soft Top-K | $0.2401 \pm 0.0175$ | $0.1415 \pm 0.0064$ | $\mathbf{0.1106 \pm 0.0035}$ | $\mathbf{0.2602 \pm 0.0185}$ | $0.1654 \pm 0.0169$ | $\mathbf{0.1301 \pm 0.0043}$ | $0.2756 \pm 0.0128$ | $0.1725 \pm 0.0098$ | $0.1329 \pm 0.0143$ |
| Policy | $\mathbf{0.237 \pm 0.0102}$ | $\mathbf{0.1326 \pm 0.0119}$ | $0.1204 \pm 0.0044$ | $0.2686 \pm 0.0263$ | $\mathbf{0.1643 \pm 0.0088}$ | $0.1543 \pm 0.0107$ | $\mathbf{0.2531 \pm 0.0078}$ | $\mathbf{0.1544 \pm 0.0053}$ | $\mathbf{0.1103 \pm 0.0105}$ |

### A.4 Discussion of Additional Results

The results on both IHDP and TCGA datasets demonstrate the effectiveness of our proposed methods in high-dimensional settings. For the IHDP dataset, we observe that the Greedy method performs particularly well for smaller acquisition sizes, while the Policy method shows strengths in larger acquisition sizes and with more experiments conducted.

On the TCGA dataset, which has a much higher dimensionality (4000 features), we see that both Soft Top-K and Policy methods outperform the random baseline consistently. The Policy method, in particular, shows robust performance across different acquisition sizes and number of experiments, highlighting its effectiveness in handling high-dimensional data.

These additional results further support our main findings that our proposed methods, especially the Policy-gradient based approach, offer robust and scalable performance on high-dimensional data, providing significant improvements over random experimentation in estimating conditional treatment effects.

## B Detailed Algorithm Descriptions

This section provides more detailed descriptions of the algorithms used in our study.

### B.1 Top-K Algorithm

The Top-K algorithm selects the top-k designs with the highest estimated Expected Information Gain (EIG) per design. Here's a more detailed description of the algorithm:

---
**Algorithm 1:** Top-K batch selection

---
1 **for** $b = 1$ **to** $B$ **do**
2 $\quad \mathbf{x}^*, \mathrm{t}^* = \arg\max_{\mathbf{x} \in D_{pool}, \mathrm{t} \in T} \mathrm{EIG}(\{(\mathbf{x}, \mathrm{t})\})$
3 $\quad$ Add $(\mathbf{x}^*, \mathrm{t}^*)$ to the batch
4 $\quad D_{pool} \leftarrow D_{pool} \setminus \{\mathbf{x}^*\}$ $\qquad\qquad\qquad$ ▷ Remove selected unit from pool
5 **end**

---

### B.2 Softmax Top-K Algorithm

The Softmax Top-K algorithm introduces stochasticity to the selection process by sampling from a softmax distribution of the EIG scores:

---
**Algorithm 2:** Softmax top-k batch selection

---
1 **for** *update step* $c = 1 \dots B$ **do**
2 $\quad \mathbf{x}^*, \mathrm{t}^* = \arg\max_{\mathbf{x} \in D_{pool}, \mathrm{t} \in T} \mathrm{EIG}(\{(\mathbf{x}, \mathrm{t})\}) + \epsilon$
$\quad\quad$ ▷ Select unit without replacement
3 $\quad D_{pool} \leftarrow D_{pool} - \mathbf{x}^*$

---

### B.3 Greedy Optimization Algorithm

The Greedy Optimization algorithm iteratively selects the experiment that maximizes the marginal gain in EIG:

---

**Algorithm 3:** $1 - \frac{1}{\epsilon}$ Greedy batch selection

---

**1** $A \leftarrow \emptyset$
**2** **for** *update step* $c = 1 \dots B$ **do**
**3** $\quad$ $\mathbf{x}^*, \mathrm{t}^* = \arg\max_{\mathbf{x} \in D_{pool}, \mathrm{t} \in T} \mathrm{EIG}(A \cup \{(\mathbf{x}, \mathrm{t})\})$
**4** $\quad$ $A \leftarrow A \cup \{\mathbf{x}^*, \mathrm{t}^*\}$
$\quad\quad$ ▷ Select unit without replacement
**5** $\quad$ $D_{pool} \leftarrow D_{pool} - \mathbf{x}^*$

---

### B.4 Policy-Gradient Based Optimization Algorithm

The Policy-Gradient Based Optimization algorithm uses gradient ascent to optimize a parameterized policy for selecting experiments:

---

**Algorithm 4:** Policy-Gradient Based Optimization batch selection

---

**1** Initialize policy parameters $\phi_x$ for units and $\phi_t$ for treatments
**2** **for** $c = 1$ **to** $C$ **do**
**3** $\quad$ Sample batch $\{(\mathbf{x}, \mathrm{t})_{1:B}\} \sim \pi_\phi(X, T)$
**4** $\quad$ Calculate $\mathrm{EIG}(\{(\mathbf{x}, \mathrm{t})_{1:B}\})$
**5** $\quad$ Update $\phi_x \leftarrow \phi_x + \alpha_x \nabla_{\phi_x} \mathrm{EIG}(\{(\mathbf{x}, \mathrm{t})_{1:B}\})$
**6** $\quad$ Update $\phi_t \leftarrow \phi_t + \alpha_t \nabla_{\phi_t} \mathrm{EIG}(\{(\mathbf{x}, \mathrm{t})_{1:B}\})$
**7** **end**
**8** Sample final batch $\{(\mathbf{x}^*, \mathrm{t}^*)_{1:B}\} \sim \pi_\phi(X, T)$

---

### B.5 Expected Information Gain (EIG) Calculation

The Expected Information Gain is calculated using a Monte Carlo estimation:

$$\mathrm{EIG}(\{(\mathbf{x}, \mathrm{t})_{1:B}\}) \approx \frac{1}{M} \sum_{m=1}^{M} \log \frac{p(y_m | \mathbf{x}, \mathrm{t}, \theta_m)}{\frac{1}{N} \sum_{n=1}^{N} p(y_m | \mathbf{x}, \mathrm{t}, \theta_n)} \tag{4}$$

where $\theta_m \sim p(\theta | \mathcal{D})$ are samples from the posterior distribution over model parameters, $y_m \sim p(y | \mathbf{x}, \mathrm{t}, \theta_m)$ are samples from the predictive distribution, and $M$ and $N$ are the number of Monte Carlo samples used.

These algorithms form the core of our approach to efficient experimentation for estimating conditional treatment effects. Each algorithm offers different trade-offs between computational complexity and the quality of selected experiments, as discussed in the main text.

## C Datasets

### C.1 Synthetic Dataset

The synthetic dataset is designed to be suitable for both discrete and continuous treatments. We develop a dose-response function based on the generalized dose response function introduced in Taleb & West (2023).

The **covariate** $x$ is generated from a normal distribution reflecting some underlying continuous variable:

$$x \sim N(70, 20)$$

**Treatments** $t$ are continuous and normally distributed:

$$t \sim N(15, 5)$$

The **outcome** $Y$ for each unit is computed through a response function that depends on both the covariate $x$ and treatment $t$. The function is defined as:

$$Y = \min(f_1, f_2)$$

where

$$f_1 = \frac{1}{1 + e^{-t+\text{offset}-5}}$$

$$f_2 = -\frac{2}{1 + e^{(-t+\text{offset}+2)\cdot 3}} + 1$$

and

$$\text{offset} = \frac{x - 20}{4}$$

We standardize the covariates, treatments, and output to enable the model to learn the function more easily.

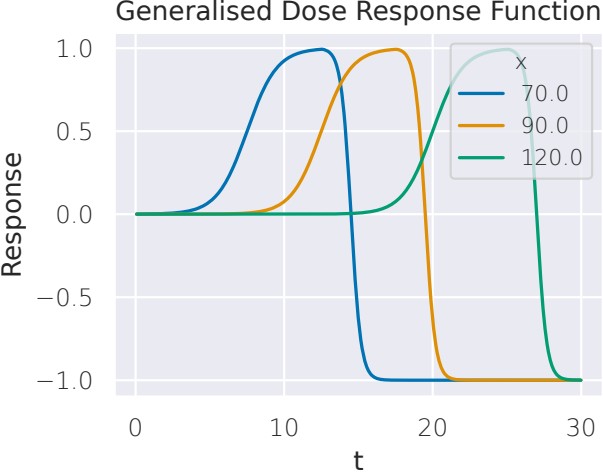

Figure 2: The dose response from which our datasets are constructed, visualized for 3 different covariate values.

For the discrete version of this dataset, we simply choose two values for $t$, 10 and 20, and use the above dose-response function.

### C.2   IHDP Semi-Synthetic Dataset

The Infant Health and Development Program (IHDP) is a semi-synthetic dataset (Shalit et al., 2017) commonly used in literature to study the performance of causal effect estimation methods. Each unit is represented by 25 covariates describing different aspects of the infants and their mothers.

The **covariate** $x$ is a normalized 25-dimensional feature vector derived from the IHDP dataset. We produce a response function from it by projecting it into a 1-D space, using the parametric Uniform Manifold Approximation and Projection (UMAP) Sainburg et al. (2020) algorithm. We then normalize it to a normal distribution reflecting the following distribution:

$$x \sim N(70, 20)$$

**Treatments** $t$ are continuous and normally distributed:

$$t \sim N(15, 5)$$

The **outcome** $Y$ for each unit is computed using the same response function as in the synthetic dataset.

This semi-synthetic dataset uses high-dimensional covariates seen by the model, but the underlying dose-response function is based on the 1-D projection.

### C.3 TCGA Semi-Synthetic Dataset

The TCGA dataset consists of gene expression measurements for cancer patients (Cancer Genome Atlas Research Network et al., 2013). There are 9659 samples for which we used the measurements from the 4000 most variable genes. The gene expression data was log-normalized and each feature was scaled in the [0, 1] interval. For each patient, the features were scaled to have norm 1. We used the same version of the TCGA dataset as used by DRNet (Schwab et al., 2020).

The construction of this semi-synthetic dataset follows the construction of the IHDP Semi-Synthetic Dataset, except for the parametric UMAP algorithm reducing the dimensionality of the covariates from 4000 to 1, as an input to the dose-response function. Again, the model sees the high-dimensional covariates.

## D   Training S-Learners

For all experiments, we use Deep Ensembles (Lakshminarayanan et al., 2017) as our model, which can be seen as an approximate Bayesian inference method (Wilson & Izmailov, 2020). The specific architectures and training details for each dataset are as follows:

### D.1   Synthetic Dataset

- MLP with 2 hidden layers and 512 hidden units
- Optimizer: Adam, with a learning rate of 0.001
- Deep ensemble of 10 MLPs
- Trained for a maximum of 200 epochs per round

### D.2   IHDP Dataset

- MLP with 2 hidden layers and 1024 hidden units
- Optimizer: Adam, with a learning rate of 0.001
- Deep ensemble of 10 MLPs
- Trained for a maximum of 200 epochs per round

### D.3   TCGA Dataset

- MLP with 3 hidden layers and 1024 hidden units
- Optimizer: Adam, with a learning rate of 0.0005
- Deep ensemble of 10 MLPs
- Trained for a maximum of 200 epochs per round

## E   Policy-Gradient Hyperparameters

The policy-gradient method requires two additional hyperparameters: learning rates for variables that describe the unit and treatment. We performed a simple sweep over the parameters and found that learning rates of 0.01 and 0.001 for the unit and treatment, respectively, are relatively robust. We optimize these parameters using Adam.

## F  Compute Used

We used an internal cluster of 24 Nvidia Titan RTX cards for our experiments. The running time of individual experiments was relatively short, typically less than 4 hours each. However, an attempt to use the greedy algorithm on the TCGA dataset exceeded 14 hours before it was cancelled due to time constraints.

In total, we estimate that across 4 datasets (synthetic discrete and continuous, IHDP, and TCGA), 4 methods, 3 acquisition sizes, and 10 seeds, approximately 480 experiments were conducted for this paper. With an average running time of 2 hours, this amounts to approximately 960 GPU-hours, along with associated CPU usage.

## G  Related Work

This section provides a more detailed overview of related work in the fields of Bayesian Optimisation, Bayesian Optimal Experimental Design, Contextual Bandits, and Active Learning.

### G.1  Bayesian Optimisation

Our setting is related to contextual Bayesian optimisation. Some of the more closely related methods include Profile Expected Improvement (Ginsbourger et al., 2014), Multi-task Thompson Sampling (Char et al., 2019), and conditional Bayesian optimization (Pearce et al., 2020). All of these methods are, however, restricted to GPs and the criteria they use to choose optimal designs are not information-based. Our work differs from this family of methods, in that we aim to estimate the entire treatment-effect relationship, rather than merely the optimal treatment. Furthermore, Bayesian Optimisation typically operates in the setting in which one can repeat experiments on the unit of interest, whereas our work assumes that each unit can only be experimented on once.

### G.2  Bayesian Optimal Experimental Design

Our objective of selecting units/treatments bears the greatest similarity to experimental design. Experimental design aims to select the most informative experiments to conduct, given a limited budget. The difference in setting between experimental design and our work is that experimental design assumes that one can repeat experiments on the same unit, whereas our work assumes that each unit can only be experimented on once. The most closely related to our EIG objective function in the context of sequential experimental design is Ivanova et al. (2022). However, Ivanova et al. (2022) are primarily interested in a secondary utility objective added in the EIG. Similar methods for obtaining variational EIG objectives have been used in implicit likelihood BED methods for *parameter learning*, but not for contextual optimisation. Gradient-based methods for large batch experimentation include SG-BOED (Foster et al., 2020) and MINEBED (Kleinegesse & Gutmann, 2020), while the policy-based iDAD (Ivanova et al., 2021) applies to batch and adaptive settings. Our ability to handle discrete designs is another important distinction of our framework.

### G.3  Contextual Bandits

Contextual bandits is another broad framework that our work is related to. An extensive line of research is focused on *online linear* bandits and discrete actions chosen using (variations of) UCB, Thompson sampling or $\epsilon$-greedy strategy (Auer, 2002; Chu et al., 2011; Agrawal & Goyal, 2013; Han et al., 2020). Krause & Ong (2011) instead model the reward as a GP defined over the context-action space and develop CGP-UCB. More recently, Zanette et al. (2021) proposed designing a batch of experiments *offline* to collect a good dataset from which to learn a policy.

### G.4  Active Learning

Deng et al. (2011) propose the use of Active Learning for recruiting patients to assign treatments that will reduce the uncertainty of an Individual Treatment Effect model. They focused on a small number of heterogeneous groups rather than individuals, with discrete treatments. As such, they tackle their setting through multi-armed bandits. While their objective remains the same as ours, in part our work could be seen as an extension of this work, with this work focusing on the individual

Table 5: Comparison between various Adaptive Trial Methods and Active Learning.

| Methods | Non-replaceable Units | High-Dimensional Covariates | Discrete Treatments | Continuous Treatments | Batch Acquisition |
|---|---|---|---|---|---|
| Deng et al. (2011) | ✓ | | ✓ | | |
| Atan et al. (2019) | ✓ | | ✓ | | ✓ |
| Gal et al. (2017) | ✓ | ✓ | | | |
| Kirsch et al. (2019) | ✓ | ✓ | | | ✓ |
| Jesson et al. (2021) | ✓ | ✓ | | | ✓ |
| **Ours** | ✓ | ✓ | ✓ | ✓ | ✓ |

level and operating in a setting with high-dimensional covariates. Atan et al. (2019) similarly focus on small heterogeneous groups with discrete treatments, and propose using Knowledge-Gradients to optimize the allocation process.

## G.5 Comparison of Methods

Table 5 provides a comparison between various Adaptive Trial Methods and Active Learning approaches, highlighting the unique features of our proposed method.

As shown in Table 5, our framework is uniquely positioned to handle non-replaceable units, high-dimensional covariates, both discrete and continuous treatments, and batch acquisition. This combination of features allows our approach to be more flexible and applicable to a wider range of experimental design scenarios compared to existing methods.

