# OpenReview forum: "Efficient Experimentation for Estimation of Continuous and Discrete Conditional Treatment Effects"
_NeurIPS.cc/2024/Workshop/BDU — NeurIPS BDU Workshop 2024 Poster_

### Official Review · Reviewer_Jivb · 2024-09-18
**Peer Review for Efficient Experimentation for Estimation of Continuous and Discrete Conditional Treatment Effects**

**Rating:** 7
**Confidence:** 3

**Review:**

## 1. Summary
The paper proposes a framework aimed at improving the efficiency of experiments designed to estimate personalized treatment effects. The authors introduce an active learning approach to select the most informative unit-treatment combinations for experimentation in settings where each experimental unit can be treated only once. The authors employ Bayesian active learning techniques and introduce both greedy and policy gradient-based optimization strategies to enhance batch experimentation. The effectiveness of their approach is demonstrated through experiments on synthetic and semi-synthetic datasets.

## 2. Strengths and Weaknesses
### Originality
The originality of this paper is evidenced through the combination of active learning and treatment effect estimation, which is not widely represented in existing literature. While relevant literature is cited adequately, further elaboration on the unique contributions compared to previous studies could make it easier for readers to understand the original contribution of this paper better.

### Quality
The paper is of good quality, as the claims are both supported by clear theoretical formulation and empirical experiments. In addition, the paper provides tabled results and illustrative visuals to document the findings.

However, there are three feedback/questions for authors' consideration -
1. Consider discussing the limitations of proposed methods. Are there any limitations of Greedy and Policy-Gradient based optimization, compared to Softmax Top-K (e.g., solve time, computational resources)?
2. Consider adding more explanation for the covariate-response curve
3. Is there a particular consideration for not including the results of Top-K method?

### Clarity
Overall, the paper is well-organized and clearly written. The structure allows readers to follow the logical progression from problem formulation to method implementation and results. Particularly, the paper summarized and discussed the key findings. Excellent work!

### Significance
While the results of this paper demonstrate improvements in experimentation efficiency, the paper could benefit from discussing more empirical significance of the findings.

It might be beneficial to add a section to discuss the potential real-world applications and implications. Which field/industry do you see this applied to?

## 3. Questions
1. What are the limitations of Greedy and Policy-Gradient based optimization, compared to Softmax Top-K (e.g., solve time, computational resources)?
2. Could the authors provide more insight into how the results would scale with larger datasets or different configurations of unit-treatment combinations?
3. How do the authors envision their methods being integrated into existing experimental designs in clinical trials or other fields?

## 4. Limitations
As discussed above, this paper could benefit from a more thorough discussion of 1) the limitations of proposed methods, 2) potential real-life applications and implications.

But all in all, the paper presents a compelling approach to improving experimentation efficiency for treatment effect estimation.

---

### Official Review · Reviewer_Hi5e · 2024-09-26
**In this paper, the authors propose four approaches for experiment design. While it's an interesting contribution, I've identified several areas for improvement. I may have overlooked something, but I'm struggling to fully grasp the paper's primary contribution. I've included my comments below, and I hope the authors find them helpful.**

**Rating:** 4
**Confidence:** 3

**Review:**

In Section G.1, the authors state that Bayesian Optimization assumes repeated experiments. However, an alternative approach would be to remove units from the pool after use, similar to the algorithms in Sections B.1, B.2, and B.3. I believe Bayesian Optimization with Gaussian Processes could be applied to this problem unless I am missing something.

To enhance the readability of the plots in Appendix A.1, please include descriptive titles and labels. This will aid in visually differentiating the various densities and make the results more interpretable.

Throughout the paper, references to "our method" and "our dataset" can be confusing. Please explicitly state the specific method and dataset being used in each context. This will help readers understand the experiments and results more clearly.

The paper mentions that policy-based algorithms optimize "joint mutual information." However, the methodology section lacks details on how this is implemented. Please provide information on your approach that incorporates joint mutual information.

Based on the results in Tables 1-4, it appears that greedy and softmax methods often perform similarly or even better than policy-based methods. This raises the question of whether policy-based sampling is truly necessary.

The paper's primary contribution seems to be the development of a method to handle both continuous and discrete treatment options. This is achieved using Bayesian Neural Networks, which, from my understanding, treat discrete variables as a special case of continuous variables. Please clarify how your proposed methodology supports discrete and continuous treatment options.

---

### Decision · Program_Chairs · 2024-10-09

Accept (Poster)